

# Clinical and prognostic value of preoperative hydronephrosis in upper tract urothelial carcinoma: a systematic review and meta-analysis

Yuejun Tian[1], Yuwen Gong[1], Yangyang Pang[1], Zhiping Wang[1] and Mei Hong[1,2]

[1] Institute of Urology, Lanzhou University Second Hospital, Lanzhou University, Lanzhou, Gansu, China
[2] Drug Discovery Center, School of chemical Biology and Biotechnology, Peking Universtiy Shenzhen Graduate School, Shenzhen, Guangdong, China

## ABSTRACT

**Background.** Epidemiological studies have reported various results relating preoperative hydronephrosis to upper tract urothelial carcinoma (UTUC). However, the clinical significance and prognostic value of preoperative hydronephrosis in UTUC remains controversial. The aim of this study was to provide a comprehensive meta-analysis of the extent of the possible association between preoperative hydronephrosis and the risk of UTUC.

**Methods.** We searched PubMed, ISI Web of Knowledge, and Embase to identify eligible studies written in English. Summary odds ratios (ORs) or hazard ratios (HRs) and 95% confidence intervals (CIs) were calculated using fixed-effects or random-effects models.

**Results.** Nineteen relevant studies, which had a total of 5,782 UTUC patients enrolled, were selected for statistical analysis. The clinicopathological and prognostic relevance of preoperative hydronephrosis was evaluated in the UTUC patients. The results showed that all tumor stages, lymph node status and tumor location, as well as the risk of cancer-specific survival (CSS), overall survival (OS), recurrence-free survival (RFS) and metastasis-free survival (MFS) were significantly different between UTUC patients with elevated preoperative hydronephrosis and those with low preoperative hydronephrosis. High preoperative hydronephrosis indicated a poor prognosis. Additionally, significant correlations between preoperative hydronephrosis and tumor grade (high grade vs. low grade) were observed in UTUC patients; however, no significant difference was observed for tumor grading (G1 vs. G2 + G3 and G1 + G2 vs. G3). In contrast, no such correlations were evident for recurrence status or gender in UTUC patients.

**Conclusions.** The results of this meta-analysis suggest that preoperative hydronephrosis is associated with increased risk and poor survival in UTUC patients. The presence of preoperative hydronephrosis plays an important role in the carcinogenesis and prognosis of UTUC.

Corresponding author
Mei Hong, meihonglzu@163.com,
meihong@pkusz.edu.cn

## INTRODUCTION

Upper tract urothelial carcinoma (UTUC), including tumors of the urothelium of the renal pelvis and the ureter, accounts for approximately 5–10% of urinary tract carcinomas (*Roupret et al.*, *2015*; *Siegel, Naishadham & Jemal*, *2012*). The gold standard management of UTUC is radical nephroureterectomy with bladder-cuff excision for adequate local tumor control and a better survival outcome (*Clark et al.*, *2013*; *Roupret et al.*, *2013*).

Numerous studies have been conducted to identify the significant prognostic factors of UTUC. The powerful prognostic factors consist of the pathological stage, tumor location, metastasis status, lymphovascular invasion, multi-focality and tumor grade (*Kikuchi et al.*, *2009*; *Ouzzane et al.*, *2011*; *Raman et al.*, *2010*; *Roscigno et al.*, *2008*; *Roupret et al.*, *2013*). Improved knowledge of the risk factors would help us make better prognostic evaluations for a more effective therapeutic strategy. However, several other putative risk factors have been proposed, and sometimes conflicting results are presented. Preoperative hydronephrosis is also a controversial risk factor. *Ito et al.* (*2011*) confirmed that preoperative hydronephrosis is an independent predictor of poor tumor pathological outcomes in a study of patients with UTUC after nephroureterectomy. In addition, *Brien et al.* (*2010*) reported that preoperative hydronephrosis can identify at-risk UTUC patients. However, controversial results have shown that there is no correlation between preoperative hydronephrosis and the UTUC pathological stage, tumor grade, recurrence, or progression and that preoperative hydronephrosis is not an independent predictor of these outcomes of patients with UTUC (*Favaretto et al.*, *2012*; *Sakano et al.*, *2015*). With regard to survival, *Morizane et al.* (*2013*) demonstrated the positive impact of preoperative hydronephrosis using univariate analysis. However, the results of preoperative hydronephrosis as an independent prognostic factor for UTUC using multivariate analyses have been discrepant. *Ito et al.*, (*2011*) have also shown that there is no statistical significance between preoperative hydronephrosis and cancer-specific survival (CSS) or metastasis-free survival (MFS) in patients with UTUC. Accordingly, here we perform a systematic review and meta-analysis to clarify whether preoperative hydronephrosis is an independent risk factor influencing the progression and survival of UTUC.

## MATERIAL AND METHODS

### Literature research

This meta-analysis followed the Preferred Reporting Items for Systematic Reviews and Meta-analyses (PRISMA) statement (*Moher et al.*, *2010*). A systematic literature search using PubMed, ISI Web of Knowledge, and Embase was conducted to retrieve clinical studies up to March 1, 2016. The search terms used included the following: "ureteral neoplasms," "urothelium," "ureter," "renal pelvis," "upper tract urothelial," "hydronephrosis," "preoperative hydronephrosis," "prognosis or prognostic or outcome," and relevant variants of these search terms. The following criteria were used to determine study eligibility: (1) must concern the connection between preoperative hydronephrosis and clinicopathological characteristics and the prognosis of UTUC; (2) UTUC patients must have been diagnosed using the standard histopathological examination criteria; and (3)

must provide information about the preoperative hydronephrosis. The exclusion criteria for the study were: (a) studies lacking original data; (b) not written in English; and (c) reviews, meta-analyses, case reports, abstracts, or meeting records. For overlapping articles, we included the most informative and latest article.

## Data extraction and quality assessment

All data were independently reviewed by two researchers (YWG and YYP) and were cross-checked. Additionally, any disagreement or uncertainty was resolved using group discussion. The quality of the selected articles was assessed according to the Newcastle-Ottawa scale (NOS) criteria (*Stang*, *2010*). For quality, scores ranged from 0 (lowest) to 9 (greatest); studies with scores of 5 or more were graded as good quality. The data extracted from these citations included the name of the first author, publication year, country, number of patients, recruitment period, sex ratio, age, cut-off, prognostic outcomes, pathological stage, lymph node dissection, tumor grade, and tumor location. The data were extracted from the original articles. Situations lacking exact data were resolved in a number of ways: multivariate outcomes were used before univariate outcomes when both were presented, but if no multivariate results were presented, univariate outcomes were used instead.

## Statistical analysis

Odds ratios (ORs) and 95% CIs were used to estimate the relationships between preoperative hydronephrosis and clinicopathological parameters, including pathologic tumor stage, tumor grade, lymph node status, tumor location, recurrence status, and gender. HRs and 95% CIs were used to evaluate the relationships between preoperative hydronephrosis and cancer-specific survival (CSS), overall survival (OS), recurrence-free survival (RFS), or metastasis-free survival (MFS). A $p$-value of $< 0.05$ was considered statistically significant. The statistical significance of the pooled ORs and HRs was evaluated using a $Z$-test. Heterogeneity among studies was evaluated using the Cochran $Q$-statistic and the $I^2$ test (*Zintzaras & Ioannidis*, *2005*). A random effects model was used when significant heterogeneity existed among studies ($P < 0.05$ or $I^2 > 50\%$); otherwise, a fixed-effects model was used. Funnel plots and Begg's test were used to evaluate the potential publication bias (*Peters et al.*, *2006*). All statistical calculations were performed using Review Manager 5.3 (The Cochrane Collaboration, Copenhagen) and STATA version 14.0 (Stata Corp, College Station, TX).

# RESULTS

## Eligible studies and quality assessment

Initially, our search strategy identified 357 articles. A total of 19 studies published between 2007 and 2016 were included in the final meta-analysis (*Bozzini et al.*, *2013*; *Chapman et al.*, *2009*; *Chen et al.*, *2013*; *Cho et al.*, *2007*; *Chung et al.*, *2014*; *Colin et al.*, *2014*; *Fradet et al.*, *2014*; *Hwang et al.*, *2013*; *Liang et al.*, *2016*; *Luo et al.*, *2013*; *Messer et al.*, *2013*; *Ng et al.*, *2011*; *Sakano et al.*, *2013*; *Xing et al.*, *2016*; *Yeh et al.*, *2015*; *Zhang et al.*, *2016*; *Zhang et al.*, *2013*; *Zhang et al.*, *2015*; *Zou et al.*, *2014*) (Fig. 1). Thirteen studies provided original information of the relationships between preoperative hydronephrosis and the

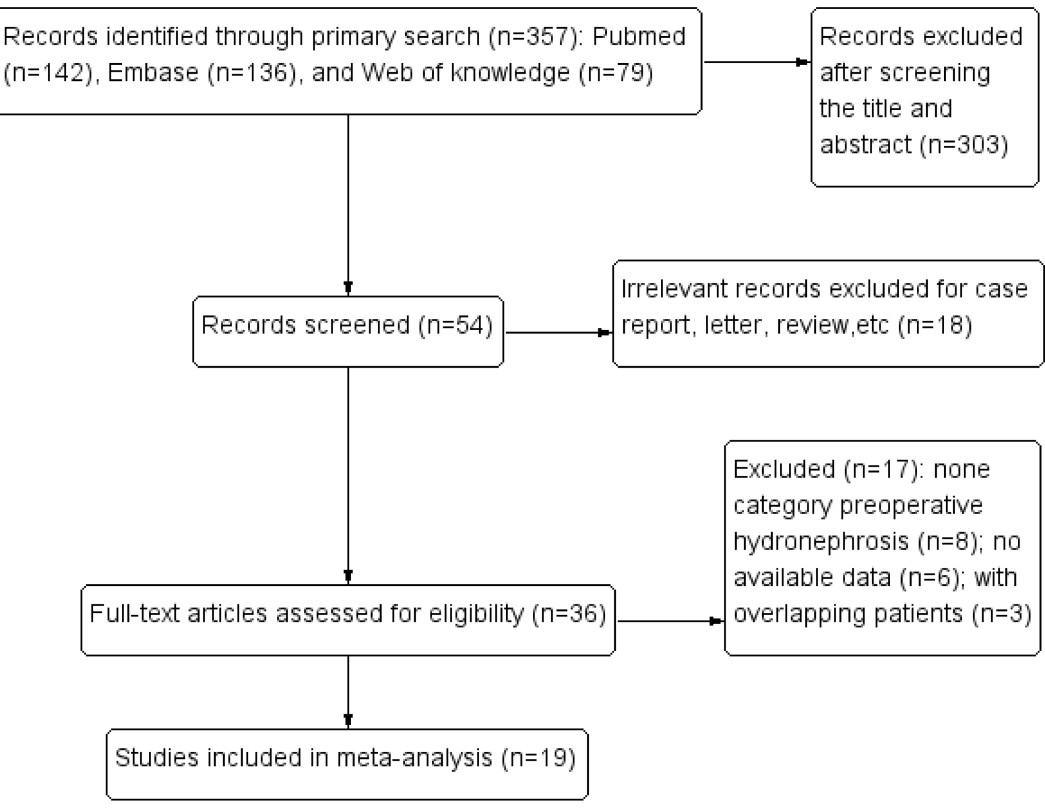

**Figure 1** Flow chart showing the study selection process.

clinicopathological parameters of UTUC patients directly (*Bozzini et al.*, *2013*; *Chapman et al.*, *2009*; *Chen et al.*, *2013*; *Cho et al.*, *2007*; *Chung et al.*, *2014*; *Fradet et al.*, *2014*; *Hwang et al.*, *2013*; *Liang et al.*, *2016*; *Messer et al.*, *2013*; *Ng et al.*, *2011*; *Yeh et al.*, *2015*; *Zhang et al.*, *2013*; *Zhang et al.*, *2015*). Fourteen articles assessed the prognostic value of preoperative hydronephrosis (CSS/OS/RFS/MFS) of UTUC patients using the Kaplan–Meier method (*Bozzini et al.*, *2013*; *Chapman et al.*, *2009*; *Chung et al.*, *2014*; *Colin et al.*, *2014*; *Hwang et al.*, *2013*; *Liang et al.*, *2016*; *Luo et al.*, *2013*; *Ng et al.*, *2011*; *Sakano et al.*, *2013*; *Xing et al.*, *2016*; *Yeh et al.*, *2015*; *Zhang et al.*, *2016*; *Zhang et al.*, *2013*; *Zhang et al.*, *2015*). The main characteristics of the 19 studies included in our meta-analysis are shown in Table 1. Other characteristics, such as the pathological results and the prognostic features, are summarized in Tables S1, S2, and S3. A total of 5,782 patients were included in this meta-analysis, and they were from 7 countries (Korea, USA, Japan, France, China, Canada, Taiwan). The median follow-up period of the studies ranged from 1 month to 233 months. The age of the patients ranged from 20 to 95 years, and the overall proportion of males was 51.74%.

Present/high preoperative hydronephrosis was defined using radiographic reports. Differences in the cut-off value of present/high preoperative hydronephrosis were observed among the studies. Preoperative hydronephrosis that was present or a high grade was considered to be positive, and absent or a low grade was considered to be negative.

**Table 1 Summary of the characteristics of enrolled studies.**

| Study | Country | Patients | Study period | Age (range), yr | Gender (m/f) | Cut off (Present/High grade) | FU (range), mon | NOS score |
|---|---|---|---|---|---|---|---|---|
| Cho_2007 | Korea | 104 | 1986–2004 | Md = 65(25–91) | 66/38 | Grade 0/1/2 vs. 3/4 on CT, EU, US | Md = 44(6–214) | 7 |
| Chapman_2009 | USA | 308 | 1996–2006 | Mn = 66.4(29.7–90.4) | 236/72 | Absence vs. presence on CT, US, MRI, IVU | NA | 8 |
| Ng_2011 | USA | 106 | 1993–2005 | Md = 69(36–90) | 67/39 | Absence vs. presence on CT | Md = 47(1–164) | 8 |
| Messer_2013 | USA | 408 | 1997–2008 | Md = 69.1(IQR, 15.5) | 254/154 | Mild/moderate vs. severe on CT, MRI, IVP, US | NA | 7 |
| Bozzini_2013 | France | 401 | 1995–2010 | Md = 69.0(IQR, 60.0-76.0) | 249/152 | Absence vs. presence on CT, MRI, IVU | Md = 26(IQR, 9.0–49.0) | 8 |
| Zhang_2013 | China | 217 | 2000–2010 | Md = 69(62–81) | 130/87 | Absence vs. presence on CT, MRI, IVP, US | Md = 52 | 8 |
| Hwang_2013 | Korea | 114 | 2004–2010 | Md = 71(41–84) | 88/26 | None/mild vs. moderate/severe on CT, EU, US | Md = 26.5 (23.5–31) | 8 |
| Luo_2013 | Taiwan | 162 | 2005–2010 | Mn = 67.97 | 81/81 | Grade 0/1/2 vs. 3/4 on urology radiologists | Md = 36.87 | 8 |
| Sakano_2013 | Japan | 536 | 1995–2009 | Md = 71(32–93) | 370/166 | Absence vs. presence | Md = 40.9(3–200) | 7 |
| Chen_2013 | China | 729 | 2002–2010 | Mn = 66.5(20–94) | 318/411 | Absence vs. presence on CT, MRI, IVU | NA | 7 |
| Zou_2014 | China | 122 | 1999–2013 | Md = 64(35–80) | 87/35 | Absence vs. presence | Md = 53(3–159) | 8 |
| Colin_2014 | France | 151 | 1995–2010 | Md = 72.5 (IQR, 63.4–78.1) | 98/53 | Absence vs. presence | Md = 18.5(IQR, 9.5–37.9) | 6 |
| Fradet _2014 | Canada | 743 | 1990–2010 | Mn = 69.7 | 438/304 | Absence vs. presence | Md = 24.8(IQR, 7.69–56.76) | 6 |
| Chung_2014 | USA | 141 | 1998–2013 | Md = 70(35–92) | 91/50 | None/mild vs. moderate/severe on CT, IVP, US | Md = 34(1–149) | 8 |
| Yeh_2015 | Taiwan | 472 | 1991–2013 | Md = 67(24–95) | 204/268 | Absence vs. presence on CT | Md = 33(1–233) | 7 |
| Zhang_2015 | China | 520 | 2000–2010 | NA | 229/291 | Absence vs. presence on CT, MRI | Md = 54(12–151) | 8 |
| Liang_2016 | China | 172 | 2001–2014 | Md = 70(IQR, 63–77) | 105/67 | None/mild vs. severe on CT, MRI, US | Md = 44(IQR, 24–62) | 7 |
| Xing_2016 | China | 192 | 2000–2013 | NA | 114/78 | Absence vs. presence on CT, US | Md = 65(3–144) | 8 |
| Zhang_2016 | China | 184 | 2006–2008 | Md = 70(61–75) | 84/100 | Absence vs. presence on CT, MRI, US | Md = 78(34–92) | 8 |

**Notes.**

Abbreviations: CT, computed tomography; EU, excretory urography; FU, follow-up; IQR, interquartile range; IVP, intravenous pyelogram; IVU, intravenous urograms; Md, median; Mn, mean; MRI, magnetic resonance imaging; NA, not available; US, ultrasound; mon, month; yr, year.

**Table 2  HR values of the CSS, OS, RFS and MFS of the UTUC.**

| Outcome | Studies (n) | Patients | HR | 95% CI | P value | Model | Heterogeneity |
|---|---|---|---|---|---|---|---|
| | | | | | | | $I^2$, P value |
| CSS | 12 | 3,063 | 1.69 | 1.23–2.33 | 0.001 | Random | 70%, 0.001 |
| OS | 6 | 1,873 | 1.62 | 1.35–1.94 | 0.000 | Fixed | 17%, 0.30 |
| RFS | 7 | 695 | 1.95 | 1.26–3.04 | 0.003 | Random | 54%, 0.04 |
| MFS | 4 | 820 | 1.55 | 1.04–2.33 | 0.03 | Fixed | 27%, 0.25 |

**Notes.**
Abbreviations: CI, confidence interval; CSS, cancer-specific survival; Fixed, fixed, inverse variance model; HR, hazard ratio; $I^2$, I-squared; MFS, metastasis-free survival; OS, overall survival; Random, random, I–V heterogeneity model; RFS, recurrence-free survival.

Fourteen articles evaluated the prognostic value of preoperative hydronephrosis (CSS/OS/RFS/MFS) in UTUC patients. Of the 14 studies, 12 provided HR and 95% CI values directly; of the other two studies, one paper provided the relative risk (RR), and the other article provided OR values, which we used to estimate HR. Of the 14 studies, a significant association between preoperative hydronephrosis and poor CSS, OS, RFS or MFS was demonstrated in six (*Chung et al.*, *2014*; *Liang et al.*, *2016*; *Ng et al.*, *2011*; *Yeh et al.*, *2015*; *Zhang et al.*, *2013*; *Zhang et al.* *2015*), four (*Chapman et al.*, *2009*; *Liang et al.*, *2016*; *Yeh et al.*, *2015*; *Zhang et al.*, *2015*), three (*Chung et al.*, *2014*; *Hwang et al.*, *2013*; *Luo et al.*, *2013*) or one (*Ng et al.*, *2011*) studies, respectively. Of the literature, the six (*Bozzini et al.*, *2013*; *Sakano et al.*, *2013*; *Xing et al.*, *2016*; *Yeh et al.*, *2015*; *Zhang et al.*, *2016*; *Zou et al.*, *2014*), two (*Bozzini et al.*, *2013*; *Yeh et al.*, *2015*), four (*Chung et al.*, *2014*; *Liang et al.*, *2016*; *Luo et al.*, *2013*; *Ng et al.*, *2011*) or three (*Bozzini et al.*, *2013*; *Colin et al.*, *2014*; *Luo et al.*, *2013*) studies linking preoperative hydronephrosis with poor CSS, OS, RFS or MFS, respectively, lacked statistical significance.

## Survival outcomes

Of the 12 studies investigating the association between preoperative hydronephrosis and CSS, the pooled HR and 95% CI for UTUC patients was 1.69 (95% CI [1.23–2.33], $P = 0.001$, $n = 3,063$) with heterogeneity ($I^2 = 70\%$, $P = 0.0001$; Table 2 and Fig. 2A). The pooled HR and 95% CI for OS provided in six studies was 1.62 (95% CI [1.35–1.94], $P < 0.00001$) with heterogeneity ($I^2 = 17\%$, $P = 0.30$; Table 2 and Fig. 2B). The results also demonstrated significant associations between presence of preoperative hydronephrosis and shorter RFS and MFS, respectively; the combined HRs were 1.95, 95% CI [1.26–3.04], $P = 0.003$ and 1.55, 95% CI [1.04–2.33], $P = 0.03$, respectively (Table 2, Figs. S3A and S3B).

## Relationships between preoperative hydronephrosis and clinicopathological parameters

In this meta-analysis, clinicopathological features such as tumor stage, tumor grade, lymph node status, tumor location, recurrence status, and gender, as impacted by the presence of preoperative hydronephrosis, were compared using the 13 studies. The results of the meta-analysis showed significant associations between presence of preoperative

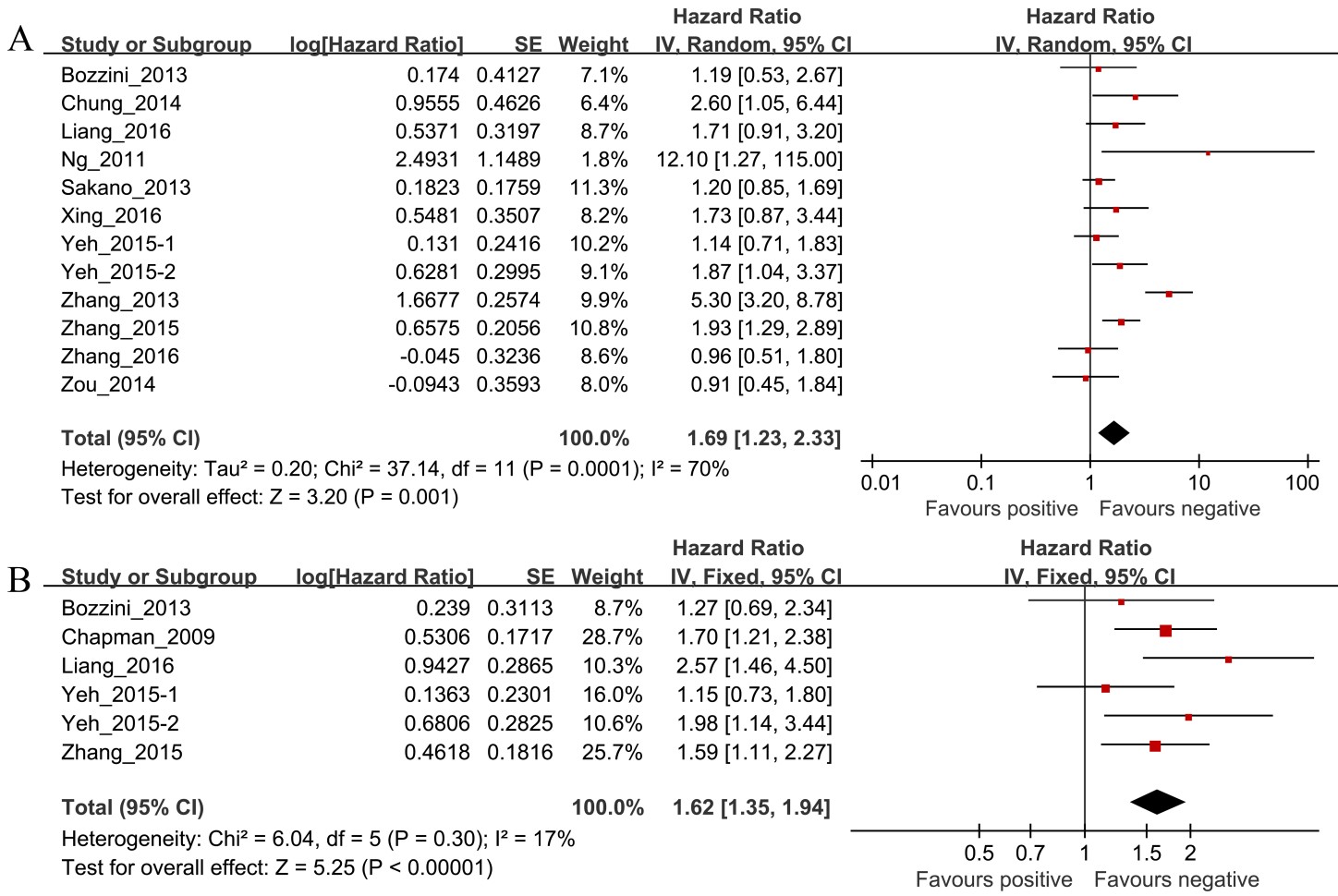

**Figure 2** (A) The hazard ratio (HR) of preoperative hydronephrosis associated with CSS in UTUC patients; (B) The hazard ratio (HR) of preoperative hydronephrosis associated with OS in UTUC patients.

hydronephrosis and higher tumor stage (T2-4), positive lymph node metastatic status or ureter tumor location, respectively; the combined ORs and 95% CIs were as follows: OR 3.12, 95% CI [1.85–5.27], $P < 0.0001$; OR 1.60, 95% CI [1.13–2.26], $P = 0.009$; OR 3.75, 95% CI [2.44–5.77], $P < 0.00001$ (Table 3, Figs. S1A, S2A and S2B). Moreover, significant associations between preoperative hydronephrosis and tumor grade (high grade vs. low grade) were observed only in UTUC patients. The OR and 95% CI was as follows: OR 1.66, 95% CI [1.20–2.29], $P = 0.002$ (Table 3 and Fig. S1B). There was no significant association between preoperative hydronephrosis and tumor grade (G3 vs. G1 + G2 and G2 + G3 vs. G1, respectively); the combined ORs and 95% CIs were OR 1.42, 95% CI [0.75–2.69], $P = 0.28$ and OR 0.94, 95% CI [0.49–1.79], $P = 0.85$ (Table 3, Figs. S1C and S1D]).

Finally, there was no significant association between preoperative hydronephrosis and recurrence status (bladder recurrence vs. no bladder recurrence) or gender (male vs. female), respectively; the combined ORs and 95% CIs were OR 1.28, 95% CI [0.92–1.77], $P = 0.14$ and OR 1.01, 95% CI [0.83–1.22], $P = 0.927$ (Table 3, Figs. S2C, and S2D).

**Table 3** OR values for the UTUC subgroups according to clinical characteristics.

| Outcome of interest | Studies | Patients | OR | 95% CI | $P$ value | Model | Heterogeneity $I^2$, $P$ value |
|---|---|---|---|---|---|---|---|
| Ta/1 vs. T2-4 | 9 | 2,462 | 3.12 | 1.85–5.27 | 0.000 | Random | 86%, 0.000 |
| High grade vs. Low grade | 3 | 799 | 1.66 | 1.20–2.29 | 0.002 | Fixed | 32%, 0.23 |
| G3 vs. G1 + G2 | 2 | 921 | 1.42 | 0.75–2.69 | 0.28 | Random | 76%, 0.04 |
| G2 + G3 vs. G1 | 2 | 921 | 0.94 | 0.49–1.79 | 0.85 | Fixed | 0%, 0.42 |
| Lymph node metastasis vs. No lymph node metastasis | 6 | 1,834 | 1.60 | 1.13–2.26 | 0.009 | Fixed | 0%, 0.55 |
| Renal pelvis vs. Ureter | 10 | 2,858 | 4.28 | 2.91–6.30 | 0.000 | Random | 78%, 0.000 |
| Recurrence vs. No recurrence | 2 | 737 | 1.28 | 0.92–1.77 | 0.14 | Fixed | 0%, 0.68 |
| Gender (Male vs. Female) | 7 | 2,556 | 1.01 | 0.86–1.19 | 0.90 | Fixed | 6%, 0.38 |

**Notes.**

Abbreviations: CI, confidence interval; Fixed, fixed, inverse variance model; $I^2$, $I$-squared; OR, odds ratio; Random, random, I–V heterogeneity model.

## Publication bias

Publication bias was assessed using Begg's test and Egger's test for asymmetry only for cancer-specific survival (CSS) of UTUC (Figs. 3A and 3B). No evidence of asymmetry was found using our funnel plot. Begg's test ($P = 0.244$) and Egger's test ($P = 0.093$) suggested that our analyses were stable.

## DISCUSSION

Increasing evidence has shown that preoperative hydronephrosis can be present in bladder tumors and UTUC. The presence of preoperative hydronephrosis in patients with bladder cancer is a predictive factor for poor pathological outcome and a poor prognosis (*Bartsch et al.*, *2007*; *Divrik et al.*, *2007*). In addition, *Hurel et al.* (*2015*) reported that hydronephrosis correlated with ureteral location, renal failure, urinary infection, positive cytology, the absence of hematuria and a fortuitous UTUC diagnosis. However, despite this finding, the relationship of preoperative hydronephrosis with UTUC outcome remains unclear, and the roles of preoperative hydronephrosis in UTUC and its clinical significance have not yet been thoroughly investigated.

The results of the analysis of the pooled data of this study showed the following: (a) preoperative hydronephrosis was associated with tumor stage, lymph node metastatic status, and tumor location in UTUC patients; (b) preoperative hydronephrosis was not strongly associated with tumor grade, gender or recurrence status in UTUC patients; (c) UTUC patients with preoperative hydronephrosis had a lower survival rate than those without preoperative hydronephrosis; (d) ureteral tumors were associated with a poorer prognosis than renal pelvic tumors (*Park et al.*, *2009*; *Park et al.*, *2004*; *Wu et al.*, *2014*). A

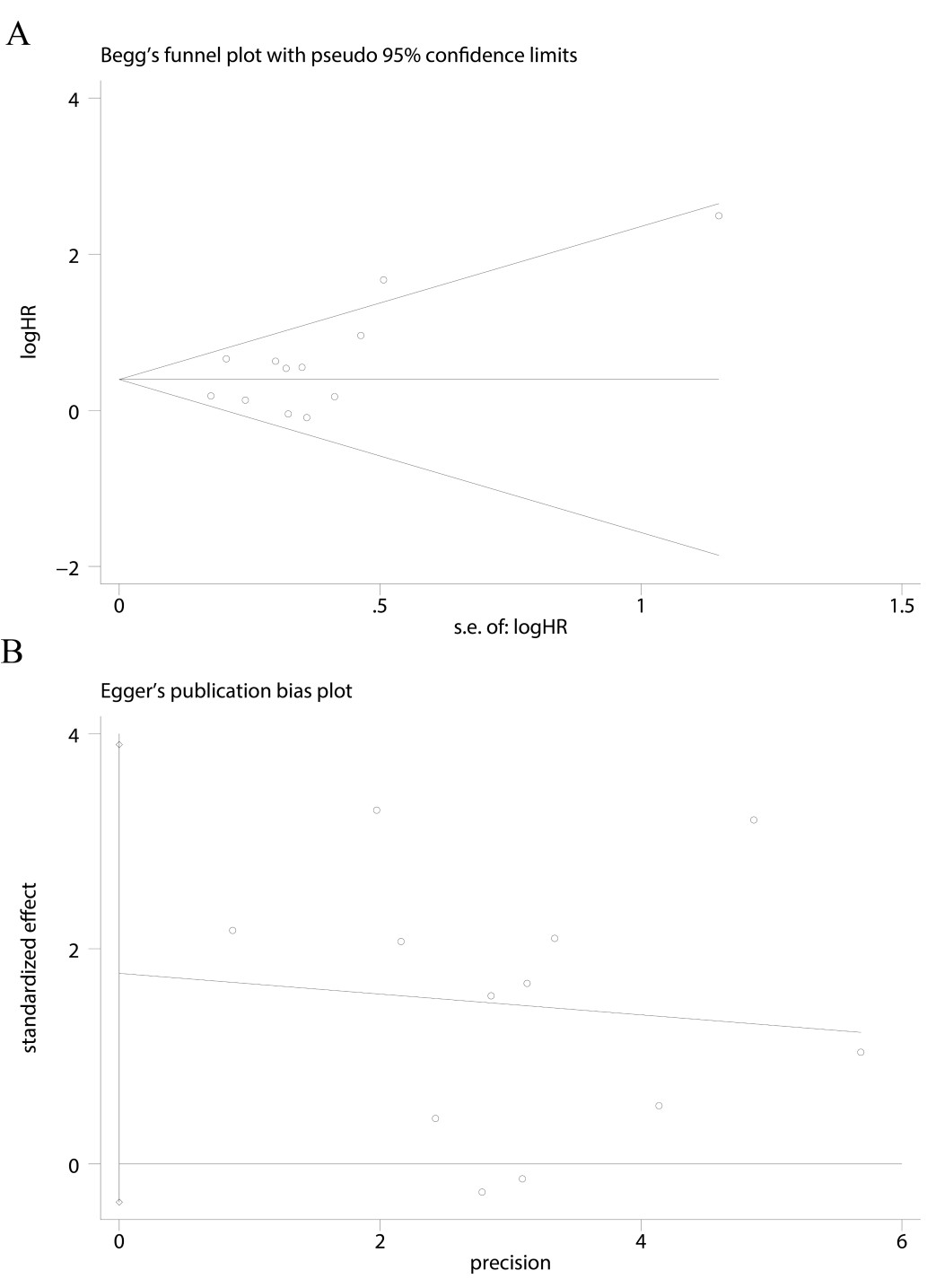

**Figure 3 Funnel plots were used to evaluate publication bias on CSS.** Begg's test and Egger's test were not significant indicating that no significant bias was observed on CSS (A and B).

hypothesis to explain this result at least partially may be that ureteral tumors are more likely to have hydronephrosis.

The biological mechanism of preoperative hydronephrosis explains its prognostic significance in UTUC. The presence of preoperative hydronephrosis is associated with poor renal function (*Hoshino et al.*, *2012*; *Rodriguez Faba et al.*, *2014*). Because long-term obstruction of the upper urinary tract would lead to renal function damage, it is not difficult to understand the relationship between hydronephrosis and renal dysfunction. *Ng et al.* (*2011*) pointed out that preoperative hydronephrosis is common in UTUC patients and may be due to one of several factors including intramural invasion, luminal obstruction and extrinsic compression. Furthermore, some researchers speculated that hydronephrosis may induce outward expansion and longitudinal thinning of the already narrow renal pelvis or ureter wall, which may promote the seeding of cancer cells to regional or distant organs. Hydronephrosis may also induce increased outward centrifugal pressure causing counter flow in lymphatics and vasculature, which may lead to increased cancer seeding (*Chung et al.*, *2014*). The presence of preoperative hydronephrosis is more common in UTUC than in bladder cancer, the reason of which may be that a small mass is more likely to cause urinary tract obstruction in the ureter (*Zhang et al.*, *2015*). In addition, *Stravodimos et al.* (*2009*), using immunohistochemical and morphological studies, found that hydronephrosis could lead to ischemic changes, along with increased expression of hypoxia-inducible factor-1a (HIF-1a), in UTUC. HIF-1a is thought to be associated with enhanced tumor cell growth and neovascularization, which is also correlated with aggressive cancer behavior (*Chai et al.*, *2008*; *Deniz et al.*, *2010*).

To our knowledge, the present meta-analysis is the first study to systematically evaluate the associations of preoperative hydronephrosis and clinicopathological features and prognostic value in UTUC. The presence of preoperative hydronephrosis predicted poorer pathological outcome and was a significant risk factor affecting survival.

Several limitations of this study need to be acknowledged. Regarding the studies included, the first limitation may be related to the prevalent adoption of retrospective studies, for the reason that almost no prospective studies were identified. Moreover, other clinical factors, such as race, age, sample size, different surgical approaches or different chemotherapies, of each study might lead to bias. Non-English studies, unpublished studies, and studies that did not provide sufficient data of the calculated HRs did not contribute to assessing the predictive value of preoperative hydronephrosis for survival. These approaches may have produced errors because of possible inaccurate reading. Finally, although we included 19 studies comprising 5,782 cases in this meta-analysis, some studies were categorized for subgroup analysis, and several survival subgroup analyses lacked data. Therefore, these results need to be further confirmed using an adequately designed prospective study to provide a better conclusion with respect to the relationship between preoperative hydronephrosis and the outcome of patients with UTUC.

## CONCLUSIONS

Although larger well-designed studies including more ethnic groups, as well as larger population studies, are required, our meta-analysis has demonstrated that preoperative

hydronephrosis was associated with prognosis-relevant factors, including tumor stage, lymph node status, and tumor location, which led to a poor CSS, OS, RFS and MFS rate in UTUC. The evaluation of preoperative hydronephrosis may therefore be informative for decisions concerning surgical strategy, and the preoperative presence of hydronephrosis should raise the possibility of employing an aggressive treatment strategy.

### Funding

This study was supported by the National Science Foundation of China (No. 81302240), the National Natural Science Foundation of China (Grant No. 81372733/H1619). The funders had no role in study design, data collection and analysis, decision to publish, or preparation of the manuscript.

### Grant Disclosures

The following grant information was disclosed by the authors:
National Science Foundation of China: 81302240.
National Natural Science Foundation of China: 81372733/H1619.

### Competing Interests

The authors declare there are no competing interests.

### Author Contributions

- Yuejun Tian conceived and designed the experiments, performed the experiments, analyzed the data, contributed reagents/materials/analysis tools, wrote the paper, prepared figures and/or tables, reviewed drafts of the paper.
- Yuwen Gong and Yangyang Pang
  conceived and designed the experiments, analyzed the data, reviewed drafts of the paper.
- Zhiping Wang performed the experiments, reviewed drafts of the paper.
- Mei Hong conceived and designed the experiments, contributed reagents/materials/analysis tools, reviewed drafts of the paper.

### Data Availability

The research in this article did not generate any raw data. Table S5 contains the raw data for Fig. 3.

### Supplemental Information

Supplemental information for this article can be found online at http://dx.doi.org/10.7717/peerj.2144#supplemental-information.

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
