# Peer review of "Clinical and prognostic value of preoperative hydronephrosis in upper tract urothelial carcinoma: a systematic review and meta-analysis"

_PeerJ, doi:10.7717/peerj.2144_

## Round 0.1 · original submission · Major Revisions

· Academic Editor

Major Revisions

The manuscript addresses an important question of potential impact of hydronephrosis on outcomes of patients with upper tract urothelial carcinoma. Despite quite thoughtful study design and analysis, there are several substantial criticisms from the reviewers that require point-by-point addressing and major revisions.

·

Basic reporting

No Comments.

Experimental design

No Comments.

Validity of the findings

No Comments.

Additional comments

The authors of this meta-analysis paper aimed to investigate the available literature for studies on the prognostic significance of hydronephrosis in patients with upper urinary tract urothelial carcinomas (UTUC). After performing natural language searches, and after applying exclusion criteria for their selection, the authors narrowed down their search to 16 publications. Based on this meta-analysis results, the authors concluded that preoperative hydronephrosis is associated with an increased risk and poorer survival in UTUC patients.
Comments to the authors:
1. The authors refer repeatedly throughout the manuscript to the grading of the urothelial carcinomas as "Fuhrman grading". This is incorrect. Fuhrman grading is a morphological grading system that applies to renal cell carcinomas, and not urothelial carcinomas. Urothelial carcinomas are morphologically graded using either a two-tiered system (low-grade vs. high-grade) or a three-tiered system, using the G1, G2 and G3 grading included in two publications reviewed in this meta-analysis. The authors should change this in the text and figures/tables.
2. The authors report a correlation between the presence of hydronephrosis and G1/G2 vs. G3 histological grade. As this three-tiered system is not universally used any longer and as the G2 category falls in part into low-grade and in part in the high-grade (in the two-tiered system) it would be interesting for the authors to present possible associations (or lack thereof) between hydronephrosis and the three different G-grades individually, as well as between G1 vs. G2/G3.
3. The authors state that "...our meta-analysis has demonstrated that preoperative hydronephrosis has a 205 detrimental effect on the clinicopathological features and prognosis of UTUC." While the association between the presence of hydronephrosis and prognosis seems sound, no cause-and-effect, or even solid pathogenetic mechanism at the base of this association has been proven yet. As such, the above statement seems quite strong and may not reflect the actual state of our understanding. Rephrasing is suggested.
4. Please provide information on how many of the studies included lacked specific survival data.
5. In lines 191-192, it is reported that " In the studies included, the definition of the cut-off value was also different." Please provide information as to the variability of these different cut-off values.
6. The authors use the following abbreviations: CSS, OS, RFS, MFS and DSS. Although spelled out later in the manuscript, they should be spelled out before their first use.
7. There is a typographical error in table 5: "Pheterogeneity". Please correct.

Reviewer 2 ·

Basic reporting

Overall, the background and motivation were reasonably clear, but see my general comment below.

There were minor issues with grammar and syntax. For example, in the first paragraph, please rephrase "including tumor" and "during the remaining upper tract". When referring to tumor stages or grades, e.g., "greater in UTUC (T2-T4) than in UTUC (Ta-T1)", do not use parentheses. In general, the writing would benefit from a careful review.

There is very little discussion about the content in Tables 1-5. Please briefly summarize any noteworthy patterns in each.

In Table 1, please indicate whether the range for patient age is for min to max or something else.

Rename "co-factors" as "covariates" in Table 2. Where possible, please report actual p-values instead of whether the analysis was "significant" or "not significant". A p-value of 0.045 is very different from <0.0001, and a p-value of 0.055 is very different from 0.85.

Consider replacing ratios in Tables 3 and 4 with figures (e.g., stratified bar plots) so that readers can more easily grasp patterns.

In Table 5, since all models are fixed effects models, consider stating this in the caption or footnote instead of listing it for each outcome and study. Does UA/MA indicate univariate or multivariate models? Define these abbreviations in a footnote. Relabel the last column as "P-value for heterogeneity".

The range of the x-axis in all figures is very wide (spanning 0.01 to 100) and makes the individual studies appear more consistent than a narrower range would.

Please define "M-H, Random", "M-H, Fixed", "IV Fixed" in the figures.

Does Fig 3E show RFS or DSS? See comment about DSS below.

Experimental design

Overall, the analysis appears to be carefully done.

My main concerns are about the selection of studies included for each endpoint and the appropriateness of aggregating results across disparate patient cohorts, clinical settings, variable definitions, and heterogeneous estimation approaches. At a minimum, challenges involved in simple meta-analysis of these studies should be acknowledged. The authors hint at this in the last paragraph of the discussion, but an expanded acknowledgment of potential inclusion and analysis biases seems necessary to not mislead readers about the robustness of the findings.

What is the difference between CSS and DSS? I thought these were variants in terminology not definition.

Please clarify whether the number of studies used in the meta-analysis for each endpoint reflects the number of studies that reported this endpoint, had non-zero events, or something else. If studies with zero events for a given endpoint were excluded, this could bias the results (Sweeting et al., Statistics in Medicine, 2004).

Validity of the findings

The final sentence in the abstract suggests causation and, while it may be true, is not directly supported by the study design. Please remove or rephrase.

Additional comments

Beyond considering preoperative hydronephrosis as a risk factor, can the authors summarize its biological implications and why it might be associated with poorer prognosis? There are two sentences that seem related to this point in the discussion, but confusingly they are presented as part of an itemized list of results of this analysis (items (f) and (g)). A standalone (and much clarified) description of what is known about the biology would go a long way to clarify the starting point of the study and to put the results in context.

---

## Round 0.2 · accepted · Accept

· Academic Editor

Accept

Thank you for your re-submission and significant improvements of the original manuscript as suggested by reviewers. I carefully reviewed all revisions and consider them to be very thorough.